# Effect of 2’-Fucosyllactose on Beige Adipocyte Formation in 3T3-L1 Adipocytes and C3H10T1/2 Cells

**DOI:** 10.3390/foods12224137

**Published:** 2023-11-15

**Authors:** Siru Chen, Yankun Fu, Tianlin Wang, Zhenglin Chen, Peijun Zhao, Xianqing Huang, Mingwu Qiao, Tiange Li, Lianjun Song

**Affiliations:** 1Henan Engineering Technology Research Center of Food Processing and Circulation Safety Control, College of Food Science and Technology, Henan Agricultural University, Zhengzhou 450002, China; cc15837026322@163.com (S.C.); chenzhenglin0108@outlook.com (Z.C.);; 2Henan Technology Innovation Center of Meat Processing and Research, College of Food Science and Technology, Henan Agricultural University, Zhengzhou 450002, China

**Keywords:** adipocytes browning, 2’-fucosyllactose, obesity, UCP1, AMPK

## Abstract

2’-Fucosyllactose (2’-FL), the functional oligosaccharide naturally present in milk, has been shown to exert health benefits. This study was aimed to investigate the effect of 2’-fucosyllactose (2’-FL) on the browning of white adipose tissue in 3T3-L1 adipocytes and C3H10T1/2 cells. The results revealed that 2’-FL decreased lipid accumulations with reduced intracellular triglyceride contents in vitro. 2’-FL intervention increased the mitochondria density and the proportion of UCP1-positive cells. The mRNA expressions of the mitochondrial biogenesis-related and browning markers (*Cox7a*, *Cyto C*, *Tfam*, *Ucp1*, *Pgc1α*, *Prdm16*, *Cidea*, *Elovl3*, *Pparα*, *CD137,* and *Tmem26*) were increased after 2’-FL intervention to some extent. Similarly, the protein expression of the browning markers, including UCP1, PGC1α, and PRDM16, was up-regulated in the 2’-FL group. Additionally, an adenosine monophosphate-activated protein kinase (AMPK) inhibitor, compound C (1 μM), significantly decreased the induction of thermogenic proteins expressions mediated by 2’-FL, indicating that the 2’-FL-enhanced beige cell formation was partially dependent on the AMPK pathway. In conclusion, 2’-FL effectively promoted the browning of white adipose in vitro.

## 1. Introduction

Obesity, a chronic disease, has become one of the most prevalent health issues that attracts global attention, leading to huge social and economic burdens [1]. A variety of reasons, including genetic factors, environmental influences, unhealthy eating habits, irregular lifestyles, and many other factors, can contribute to overweight and obesity [2]. Obesity occurs when energy intake exceeds energy expenditure, resulting in the excessive accumulation of body fat and weight gain [3]. It has been reported that obesity is also the most significant cause of insulin resistance, hyperlipidemia and other metabolic disorders, and even different types of cancer [4,5]. The prevention and treatment of obesity are mainly divided into three intervention categories—drug, surgical, and behavioral interventions. Among them, drug intervention is the most common despite exerting different degrees of side effects on the human body, such as inducing heart valve damage and hypertension [6]. Therefore, exploring strategies for preventing and combating obesity and related diseases has become a critical challenge worldwide [7]. Food-derived active substances have attracted considerable attention for their safety and efficacy. Numerous studies have investigated the anti-obesity activities of food-derived active substances, which have emerged as a research hotspot in the field of food nutrition and health.

Further, mammalian adipose tissues (AT) can be divided into two types: white AT (WAT) and brown AT (BAT) [8]. WAT is mainly distributed in the lower skin and around the internal organs, whereas BAT is found in the interscapular, axillary, and paravertebral organs [9]. The main function of WAT is to store excess energy as triacylglycerol, whereas that of BAT is to produce heat through burning substrates, such as fatty acids and glucose, to expend energy (this is known as the process of adaptive non-shivering thermogenesis) [10]. Uncoupling protein 1 (UCP1), which is distributed in the inner membrane of the mitochondria and mainly regulated via the adrenergic signal of the sympathetic nerve, can activate the thermogenic process of brown adipocytes [11]. UCP1 can eliminate the difference in the transmembrane proton concentrations on both sides of the inner mitochondrial membrane, allowing the uncoupling of ATP synthesis with oxidative phosphorylation (OXPHOS) and the dissipation of energy from the body as heat [12]. Under certain stimuli, such as physiological stress (e.g., chronic cold exposure), cellular/growth factors, or pharmacological treatments (e.g., PPARγ agonists or β-adrenergic stimulation) [10], some WATs can exhibit BAT characteristics with high UCP1 expressions and increased mitochondrial contents, and this process can be identified as the browning of WAT [13,14]. These inducible or regenerative adipocytes are known as beige/brite adipocytes, which exhibit the activation functions of BAT and are thermogenically competent [15]. Therefore, promoting WAT browning exhibits considerable therapeutic potential for augmenting energy expenditure in the treatment of metabolic diseases [16,17].

Human milk oligosaccharides (HMOs) are one of the abundant components of breast milk, which provide important health benefits and physiological functions due to their specific structural characteristics [18]. 2’-fucosyllactose (2’-FL) is the major fucosylated HMO that makes up around 30% of the total HMOs [19]. As a prebiotic, 2’-FL can improve immune system development and prevent infectious intestinal, respiratory, and urinary diseases through enriching the growth of beneficial bacteria as well as inhibiting undesirable bacteria or pathogens [20]. Further, studies have confirmed that 2’-FL supplementation can improve the intestinal barrier integrity of diet-induced obese mice and reduce their body weight and fat mass. [21]. Moreover, 2’-FL treatment with a high-fat diet (HFD) for 28 weeks improved intestinal permeability and regulated lipid metabolism through inducing catabolism and inactivating lipogenesis in Ldlr-/-.Leiden mice [22]. Although studies have revealed that supplementation with 2’-FL can regulate intestinal flora and lipid metabolism, it is unclear whether 2’-FL can alleviate obesity through promoting the browning of WAT. Thus, the aim of this study was to explore the effect and mechanism of 2’-FL on the formation of beige adipocytes in 3T3-L1 adipocytes and C3H10T1/2 cells.

## 2. Materials and Methods

### 2.1. Materials

The 3T3-L1 pre-adipocytes and C3H10T1/2 multipotent stem cells used in this study were purchased from American Type Culture Collection (ATCC, Manassas, VA, USA). The 2’-FL (purity level ≥ 85%) was obtained from Royal DSM (Netherlands, Hague). Triglyceride (TG) detection kits, dexamethasone, insulin, and 3-isobutyl-1-methylxanthine were purchased from Solarbio Science & Technology Co., Ltd. (Beijing, China). The indomethacin used was purchased from Aladdin Reagent Co. Ltd. (Shanghai, China). The 3,3′,5-Triiodo-L-thyronine (T3) was purchased from TCI chemicals (Shanghai, China). The 3-(4,5-dimethyl-2-thiazolyl)-2,5-diphenyl-2H-tetrazolium bromide (MTT) was provided by Sigma-Aldrich (St. Louis, MO, USA). MitoTracker^®^ Deep Red FM staining reagent and lysis buffer was purchased from Beyotime Biotech. Inc. (Shanghai, China). Antibodies against UCP1 (#ab10983) and PRDM16 (#ab106410) were obtained from Abcam (Cambridge, MA, USA). Antibodies against PGC1α (T56630S), SIRT1 (PQA2569S) were obtained from Abmart (Shanghai, China). Antibodies against AMPK (#5832S) and p-AMPK (#2535S) were purchased from Cell Signaling Technology (Beverly, MA, USA).

### 2.2. Cell Culture

The 3T3-L1 pre-adipocytes and C3H10T1/2 mesenchymal stem cells were cultured in a growth medium comprising Dulbecco’s Modified Eagle’s Medium (Solarbio, Beijing, China), 10% fetal bovine serum (BioInd, Kibbutz Beit, IL, USA), and 100 μg/mL penicillin–streptomycin (HyClone, UT, USA) at 37 °C in a 5% CO_2_ humidified incubator. When the cell density reached 70–80%, the cells were cultured for another two days to make contact inhibition. After that, differentiation was induced in the cells via supplementation with an adipogenic cocktail including insulin at 10 μg/mL, dexamethasone at 1.0 μM, 3-isobutyl-1-methylxanthine at 0.5 mM, T3 at 10 nM, and indomethacin at 0.125 mM. After incubation for 2 days, growth medium was treated to cells through adding a mixed inducer (insulin (10 μg/mL) and T3 (10 nM)) for another 6 days. In the process of differentiation, the fresh medium was replaced every two days with the addition of 2’-FL.

### 2.3. MTT Assay

The cell viability was measured via MTT assay. The cells were cultured in DMEM in 96-well plates and then stimulated with different concentrations of 2’-FL (0.01–8.0 mg/mL). After incubation for 24 h, MTT at 5 mg/mL dissolved in phosphate-buffered saline (PBS) was added to the medium and reacted with cells for 4 h. Next, the medium was discarded, and 150 μL of dimethyl sulfoxide (DMSO) solution was added to each well to make the crystal of formazan crystals fully dissolved. The solution was fully mixed, and the absorbance value was detected at 550 nm.

### 2.4. Oil Red O Staining

The differentiated cells were washed twice with PBS, added to a 10% polyformaldehyde solution, and fixed for 10 min at room temperature. After that, residual polyformaldehyde was washed with PBS 2–3 times, and 1 mL of freshly prepared Oil Red O solution was added to the cells and incubated for 20 min away from light. Then, the Oil Red O staining solution was discarded, and the cells were washed with PBS. The stained lipid droplets were visualized, and the images were captured using a light microscope (Dilunguangxue, DXY-2, Shanghai, China). Next, Oil Red O was dissolved in isopropyl alcohol at 37 °C for 10 min. The solution was transferred into a 96-well culture plate and placed in a microplate reader (Model wave xs2, Bio Tek, VT, USA) to detect the change of absorbance at 510 nm.

### 2.5. Triglyceride Content Analysis

After cell differentiation for eight days, the triglyceride (TG) levels of cells were detected using commercial TG content detection kits (Solarbio, Beijing, China), following the instructions. The absorbance of the solution was measured using a microplate reader (Model wave xs2, Bio Tek, VT, USA) at 420 nm.

### 2.6. Mitochondrial Content Analysis and UCP1 Immunofluorescence

For mitochondrial content analysis, after adipocyte differentiation to the 8th day, the stock solution of MitoTracker^®^ Deep Red FM was diluted to 200 nM to prepare the working solution. A preheated MitoTracker^®^ Deep Red FM solution was treated to cells and incubated for 30 min at 37 °C. After staining, the cells were fixed with 4% paraformaldehyde for 15 min followed by washing with PBS. A Hoechst 33,342 staining solution was added to the cells and incubated at 37 °C for 10 min to stain the nucleus. For UCP1 stanning, the cell medium was discarded, washed twice with PBS, added to 4% paraformaldehyde, and fixed at room temperature for 1 h. Then, following three rinses with PBS, 0.2% TritonX-100 was added to permeate the cell membrane. Then, the cells were treated with blocking solution (PBS made up of 1% BSA) and incubated at room temperature for 1 h. After that, anti-UCP1 antibody (1:500; ab10983; Abcam, Cambridge, MA, USA) was added and incubated overnight at 4 °C. After that, after washing 3 times with PBS, adipocytes were treated with Alexa Fluor^®^ 594-AffiniPure Donkey Anti-Rabbit IgG (H + L) (1:500) for 1 h away from light. After washing with PBS 2–3 times, the cells were observed under a laser confocal microscope.

### 2.7. Real-Time qPCR Analysis

After cell differentiation for eight days, the total RNA was extracted using a total RNA isolation kit (Tiangen, Beijing, China) and quantified with a NanoDrop ND-2000 spectrophotometer (Thermo Fisher Scientific, Waltham, MA, USA). Subsequently, RNA was reverse transcribed into cDNA using a PrimeScript RT reagent kit (Tiangen, Beijing, China). Additionally, cDNA was used as a template for RT-qPCR, which was performed in a PCR system (Thermo Fisher, MA, USA). The utilized primer sequences for PCR are listed in Table 1. The RT-PCR program proceeded as follows: 95 °C for 5 min, 40 cycles of 60 °C for 20 s, and 95 °C for 15 s.

### 2.8. Western Blot Analysis

The total protein of adipocytes was extracted using a RIPA Lysis Buffer (Beyotime, Shanghai, China) containing inhibitor cocktails (protease and phosphatase). Then, the solution was centrifuged at 4 °C and 12,000 r/min for 15 min, and the supernatant was collected for the determination of protein concentrations via the BCA protein assay kit (Beyotime, Shanghai, China). Protein samples (10 μg) were separated using 10–15% SDS-polyacrylamide gels and then transferred electrophoretically onto polyvinylidene fluoride (PDVF) membranes (Millipore Corporation, Billerica, MA, USA). The membrane was blocked with TBST containing 5% skimmed milk for 1–2 h and then immunoblotted overnight at 4 °C using primary antibodies against UCP1, PGC1α, PRDM16, AMPK, p-AMPK, and SIRT1, respectively. After washing with TBST buffer 5 times, the membranes were incubated with the corresponding horseradish peroxidase (HRP)-conjugated antibodies (Beyotime, Shanghai, China) for 1 h. After washing five times with the TBST buffer, immunodetection was carried out using an enhanced chemiluminescence detection reagent. Immunoreactive bands were quantified and analyzed using Image J software, version k1.47 (NIH, Bethesda, MD, USA).

### 2.9. Statistical Analysis

The experimental data were processed with SPSS 26.0 software (Chicago, IL, USA). The results were expressed as mean ± standard deviation (SD). Statistical analysis was conducted using one-way ANOVA followed by Tukey’s test. Statistical significance was considered at *p* < 0.05. All experiments were conducted independently and repeated at least three times.

## 3. Results

### 3.1. Effects of 2’-FL on Cell Viability in 3T3-L1 Adipocytes and C3H10T1/2 Cells

As shown in Figure 1a, 2’-FL (0.01–4.0 mg/mL) showed no toxicity in the 3T3-L1 adipocytes. However, after incubation for 24 h, treatment with 2’-FL at 8.0 mg/mL decreased the cell viability. In C3H10T1/2 cells, 2’-FL treatment at 0.01–4.0 mg/mL did not affect the cell viability and had no difference compared to the control cells, whereas the 8.0 mg/mL 2’-FL treatment decreased the cell viability after 24 h of incubation (Figure 1b). Therefore, 0.5 and 2.0 mg/mL 2’-FL were confirmed to be used for the subsequent analysis.

### 3.2. 2’-FL Inhibited Lipid Accumulation in 3T3-L1 Adipocytes and C3H10T1/2 Cells

To investigate the effects of 2’-FL on adipogenesis in 3T3-L1 adipocytes and C3H10T1/2 cells, the formation of lipid droplets and the change of TG content were measured. On Day 8 of differentiation, the 2’-FL intervention group at 2.0 mg/mL showed smaller lipid droplets with reduced intracellular lipid accumulation compared to those of the control group in 3T3-L1 adipocytes (Figure 2a,b). In C3H10T1/2 cells, 2’-FL at 0.5 and 2.0 mg/mL also significantly reduced lipid droplets size and lipid accumulation compared to the control group (Figure 2c,d). Moreover, as shown in Figure 2e,f, under 2’-FL intervention, the TG contents of the 3T3-L1 adipocytes and C3H10T1/2 cells exhibited lower levels compared with the control group. These results indicate that 2’-FL inhibited lipid accumulation in 3T3-L1 adipocytes and C3H10T1/2 cells.

### 3.3. 2’-FL Induced Mitochondrial Biogenesis in 3T3-L1 Adipocytes and C3H10T1/2 Cells

To explore the effects of 2’-FL on mitochondrial biogenesis, the mitochondria were stained with MitoTracker Red to assess the mitochondrial intensity, and the mRNA expression of mitochondrial biogenesis-related genes was also tested. Compared with the control cells, treatment of 2’-FL at 0.5 and 2.0 mg/mL significantly increased the red fluorescence intensity in both 3T3-L1 adipocytes (Figure 3a) and C3H10T1/2 cells (Figure 3b), indicating that 2’-FL intervention increased the mitochondrial density. Moreover, 2’-FL at 2.0 mg/mL upregulated the mRNA expression of mitochondrial biogenesis-associated genes including *Cox7 a*, *Cyt C*, and *Tfam* in 3T3-L1 adipocytes (Figure 3e). Additionally, the mRNA expression of *Cox7 a* and *Tfam* was increased after the treatment with 2.0 mg/mL 2’-FL in C3H10T1/2 cells (Figure 3f). These results indicated that 2’-FL could promote mitochondrial biogenesis in vitro.

### 3.4. 2’-FL Increased the mRNA and Protein Expression of Key Browning Markers in 3T3-L1 Adipocytes and C3H10T1/2 Cells

To investigate the potential role of 2’-FL in promoting WAT browning, the expression of browning-related markers and thermogenic genes was determined via RT-PCR and Western blot. As shown in (Figure 3a–d), 2’-FL treatments at 0.5 and 2.0 mg/mL all significantly increased the green fluorescence intensity, which showed increased UCP1-positive adipocytes in 3T3-L1 adipocytes and C3H10T1/2 cells. Consistent with this, as shown in Figure 4a, 2’-FL treatment upregulated the mRNA expressions of browning markers including *Ucp1*, *Pgc1α*, *Prdm16*, *Cidea*, *Elovl3*, *Tmem26*, and *Cd137* in 3T3-L1 adipocytes. The gene levels of *Ucp1*, *Pgc1α*, *Pparα*, *Cidea*, *Elovl3*, *Tmem26*, and *Cd137* in the 2’-FL-treated group were higher compared to those in the control group in C3H10T1/2 cells (Figure 4b). 2’-FL at 2.0 mg/mL significantly increased the protein expressions of UCP1, PGC1α, and PRDM16 in both 3T3-L1 adipocytes (Figure 4c) and C3H10T1/2 cells (Figure 4d), whereas the protein expression of SIRT1 was only upregulated after 2’-FL intervention in C3H10T1/2 cells. Furthermore, compared with the control group, the 2’-FL treatment group exhibited an increased p-AMPK/AMPK ratio and upregulated phosphorylation of AMPK in 3T3-L1 adipocytes (Figure 4c) and C3H10T1/2 cells (Figure 4d). These results illustrate that 2’-FL can stimulate the browning of 3T3-L1 or C3H10T1/2 cells.

### 3.5. Effect of Compound C on 2’-FL -Induced Thermogenesis in 3T3-L1 and C3H10T1/2 Adipocytes

To investigate whether the effects of 2’-FL on beige cell transformation were related to the AMPK pathway, an AMPK inhibitor, compound C, was used. The 2’-FL treatment alone significantly upregulated the protein expression of UCP1, PGC1α, and PRDM16, as well as increased the ratio of p-AMPK/AMPK in 3T3-L1 adipocytes and C3H10T1/2 cells compared to the control group. However, in the presence of compound C, the increase in the protein expression of UCP1, PGC1α, and PRDM16 and the ratio of p-AMPK/AMPK induced by 2’-FL treatment were all significantly inhibited in 3T3-L1 adipocytes (Figure 5a) and C3H10T1/2 cells (Figure 5b). These findings indicated that 2’-FL mediated thermogenic effects on 3T3-L1 adipocytes and C3H10T1/2 cells was associated with the activation of the AMPK signaling pathway.

## 4. Discussion

Obesity and its associated diseases have become an epidemic crisis with increasing global prevalence. The main function of white adipocytes is to store energy, while beige/brite or brown adipocytes can generate heat. The excessive storage of WAT and loss of BAT may increase the incidence of overweight and obesity. Therefore, investigating the regulatory mechanisms of adipocyte browning will provide a new approach into the prevention and treatment of obesity and other metabolic disturbances. Inducing white adipocyte browning and promoting the formation of beige/brite adipocytes are promising strategies for alleviating obesity as they can increase the body’s energy expenditure.

In our study, 3T3-L1 pre-adipocytes and C3H10T1/2 mesenchymal stem cells were used to investigate the function of 2′-FL in vitro. The 3T3-L1 cell line is one of the most widely characterized and extensively studied cell models for preadipocyte differentiation [23], which is derived from disaggregated mouse embryo [24]. C3H10T1/2 cells with multipotent differentiation capabilities are known to differentiate into osteoblasts, chondrocytes, and adipocytes in response to certain growth factors [25]. Therefore, it is a valuable model system for analyzing the commitment of mesenchymal stem cells to the adipocyte lineage [25]. Obesity leads to excessive expansion of WAT, manifested by lipid accumulation and the enlargement of lipid droplets [26]. The browning of adipocytes can be accompanied by a decrease in lipid accumulation [27]. Several “browning agents”, such as resveratrol, ginsenoside, and Zeaxanthin, have shown the ability to reduce lipid accumulation with increased small lipid droplets in 3T3-L1 cells, implying that these adipocytes had some features of beige adipocytes [28,29,30]. Additionally, Raffinose, an oligosaccharide isolated from Costus speciosus, can alleviate lipid accumulation through attenuating lipid synthesis in differentiated HepG2 and 3T3-L1 cells [31]. Chitosan oligosaccharides can inhibit adipogenesis with decreased lipid accumulation in adipocytes and SD rats, showing anti-obesity activity [32]. A previous study also showed that 2’-FL can specifically decrease the accumulation of lipids in the liver, leading to the attenuation of liver steatosis in obese mice [22]. Consistent with these results, compared with the control group, the 2’-FL-treated group showed a lower level of lipid accumulation and had smaller lipid droplets in both 3T3-L1 adipocytes and C3H10T1/2 cells, suggesting the potential role of 2’-FL in the regulation of adipogenesis.

Mitochondria are key factors in the regulation of energy homeostasis, which can control different types of organs and tissues, including adipose tissue [33]. A growing body of research has shown a significant relevance between mitochondrial dysfunction and the development of obesity [34]. The formation of beige cells is accompanied by an increase in mitochondria density, which can produce heat through breaking down lipids through UCP1-mediated thermogenesis [35]. Many dietary compounds have the potential against obesity through increasing mitochondrial DNA numbers and inducing mitochondrial biogenesis, thus improving the beige cell differentiation to maintain metabolic homeostasis [33]. Mitochondrial biogenesis is regulated by several proteins encoded by nuclear and mitochondrial genomes, such as *Tfam* and *Cox7a* [36]. Reports showed that agaropectin-derived oligosaccharides can be localized in mitochondria to improve the function of mitochondria, leading to the amelioration of insulin resistance and metabolic disorder in HepG2 cells [37]. Alginate oligosaccharide can improve the mitochondria function of aging mice through enhancing mitochondrial biogenesis, maintaining mitochondrial integrity as well as inhibiting mitochondria from being destroyed [38]. Additionally, other kinds of milk oligosaccharide, including 3’-sialyllactose and 6’-sialyllactose, can cause the adaptation of muscle mitochondria with increased oxygen consumption to improve exercise performance [39]. Similarly, in our study, the mitochondrial intensity and the mRNA level of mitochondrial biogenesis markers were significantly increased after 2’-FL treatments.

AMPK is a general cellular energy regulator that is necessary for several processes in adipose tissue, such as mitochondrial function, energy metabolism, and beige/brown adipogenesis [40,41,42]. The induction of AMPK contributes to the emergence of beige adipocytes in WAT through UCP1-independent thermogenesis, which protects against diet-induced obesity [43]. SIRT1, a key player in glucose and fat metabolism, also can regulates the expression of PGC1α [44]. As an energy sensing network, the phosphorylation of AMPK enhances the activity of PGC1α; SIRT1 deacetylates PGC1α and induces the co-activation of its target transcription factors, thereby promoting heat production and energy expenditure [45]. However, in our study, 2’-FL increased both the phosphorylation of AMPK and the level of SIRT1 in C3H10T1/2 cells, whereas the expression of SIRT1 was not changed in 3T3-L1 adipocytes. Since AMPK activation also can inhibit the mTORC1 activation or ER stress response to regulate metabolism [46], the above result imply that the activation of beige adipocytes induced by 2’-FL may be caused by different molecular mechanisms between the two cell lines, which need be further explored. Multiple studies have shown that a variety of oligosaccharides, such as ulvan oligosaccharide and chitooligosaccharide, can regulate thermogenesis and attenuate obesity-related disorders through activating AMPK [47,48]. Additionally, 2’-FL has been reported to regulate the AMPK/SIRT1 pathway to ameliorate oxidative stress damage in the aging process of mice [49]. 2’-FL can also activate the AMPK pathway to induce autophagy, thus promoting melanin degradation [50]. Therefore, these findings further support our results that 2’-FL stimulated the thermogenesis process in 3T3-L1 adipocytes and C3H10T1/2 cells partly through AMPK pathway. However, this study only demonstrated the effect of 2’-FL on adipocyte browning in vitro, and future studies need to further verify and evaluate the effect and mechanism of 2’-FL in vivo.

## 5. Conclusions

In summary, after differentiation, 2’-FL notably alleviated lipid accumulation with smaller lipid droplets and more UCP1-positive cells in both 3T3-L1 adipocytes and C3H10T1/2 cells. 2’-FL enhanced mitochondrial abundance, as well as increased the mRNA expression of mitochondrial biogenesis-related genes. Additionally, the mRNA and protein expression of UCP1, PGC1α, PRDM16, and other thermogenic markers was upregulated after 2’-FL administration. And the thermogenic activity of 2’-FL is closely related to the activation of AMPK pathway. Taken together, our results indicate that 2’-FL can induce the formation of beige adipocytes. And 2’-FL could be applied as a functional food ingredient for preventing obesity and related metabolic disorders.

## Figures and Tables

**Figure 1 foods-12-04137-f001:**
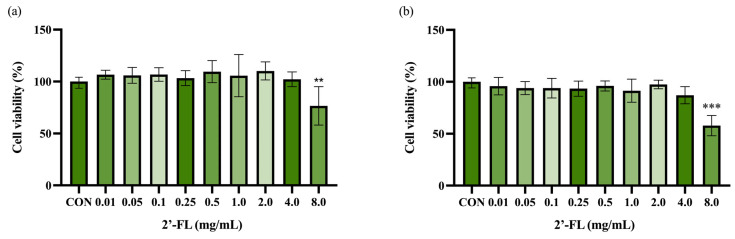
Effects of 2’-FL on the cell viability of 3T3-L1 adipocytes (**a**) and C3H10T1/2 cells (**b**). The values are expressed as ± SD (*n* = 6). ** *p* < 0.01 and *** *p* < 0.001 vs. the control group.

**Figure 2 foods-12-04137-f002:**
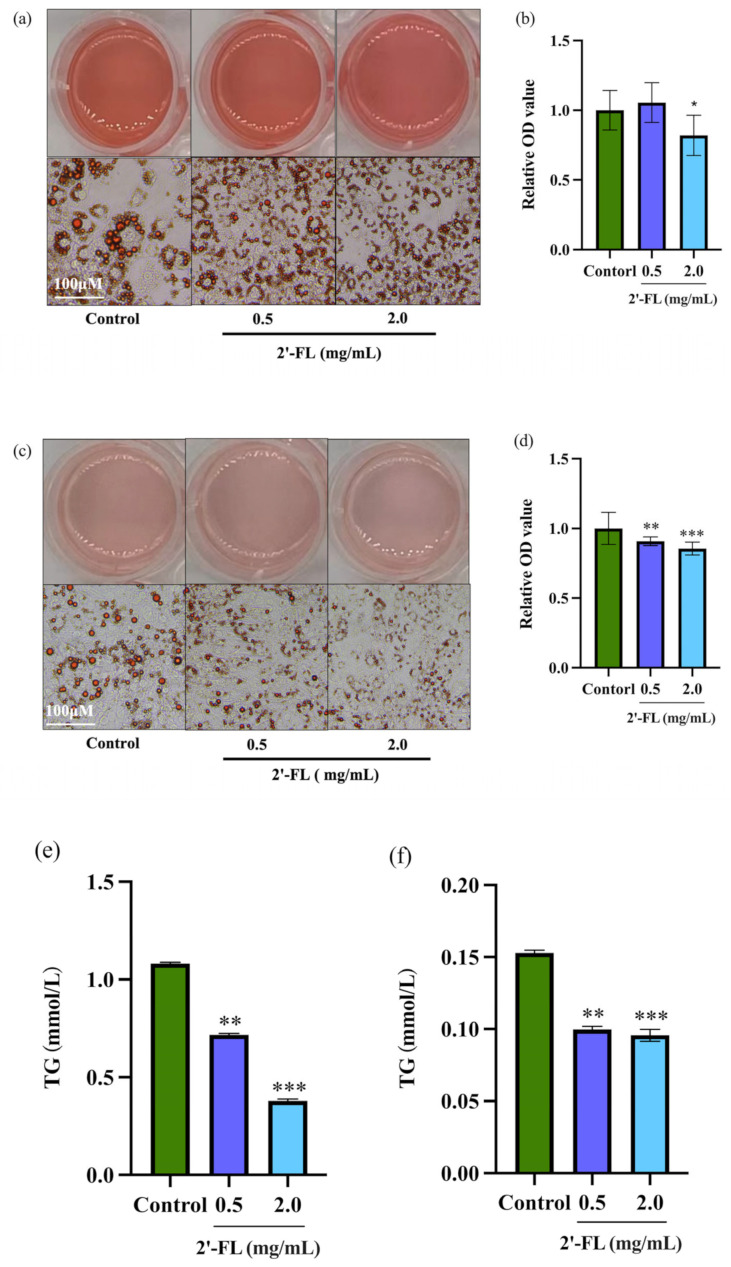
The effect of 2’-FL on intracellular lipid accumulation in 3T3-L1 adipocytes and C3H10T1/2 cells. Representative images of the 3T3-L1 adipocytes (**a**) and C3H10T1/2 cells (**c**) were observed using an optical microscope after staining with Oil Red O (Scale bar: 100 μM). Semi-quantitative analysis of the Oil Red O staining of the 3T3-L1 adipocytes (**b**) and C3H10T1/2 cells (**d**) using a microplate reader at 510 nm. (**e**,**f**) TG content measured with TG detection kits. The values are expressed as ± SD (*n* = 6). * *p* < 0.05, ** *p* < 0.01, and *** *p* < 0.001 vs. the control group.

**Figure 3 foods-12-04137-f003:**
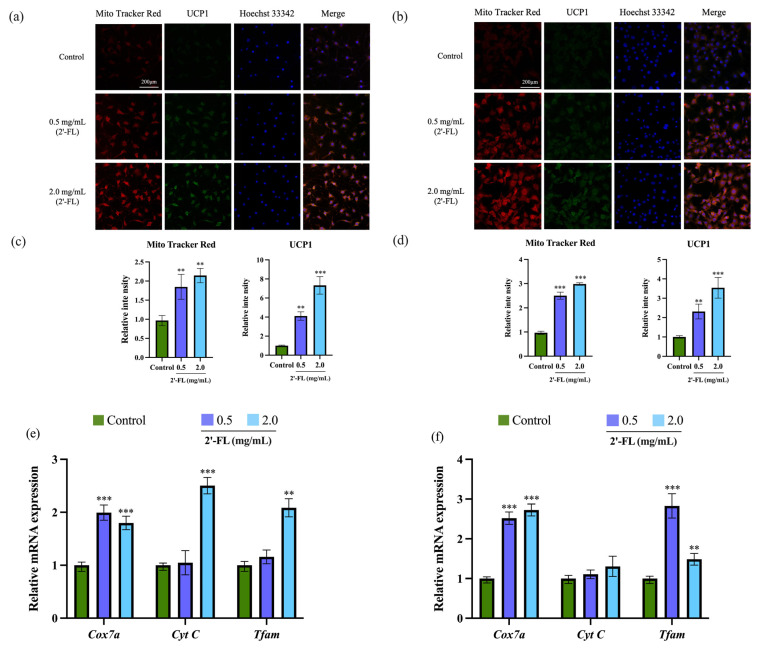
The effect of 2’-FL on the mitochondria intensity in 3T3-L1 adipocytes and C3H10T1/2 cells. The 3T3-L1 adipocytes (**a**) and C3H10T1/2 cells (**b**) were stained with MitoTracker Red and immunostained with an anti-UCP1 antibody. The images were captured using a fluorescence microscope at a ×200 magnification. The quantitative data for staining intensity of MitoTracker Red and UCP1 immunofluorescence in 3T3-L1 adipocytes (**c**) and C3H10T1/2 cells (**d**), respectively. The mRNA expression of mitochondrial biogenesis-related genes in the 3T3-L1 adipocytes (**e**) and C3H10T1/2 cells (**f**). The values are expressed as ± SD (*n* = 4). ** *p* < 0.01, and *** *p* < 0.001 vs. the control group.

**Figure 4 foods-12-04137-f004:**
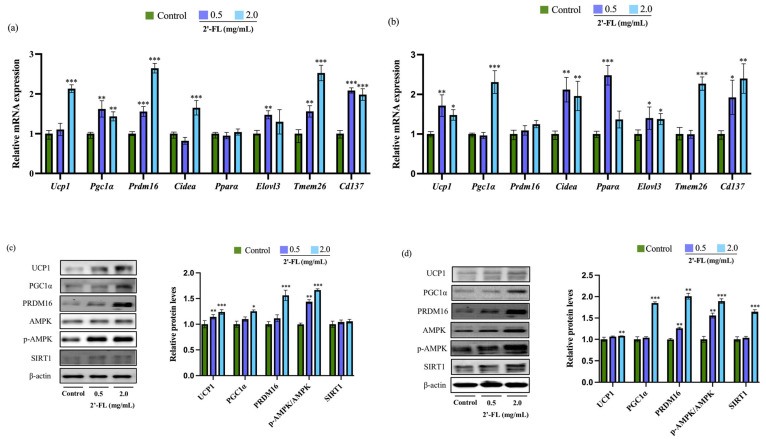
The effect of 2’-FL on the expression of beige markers and thermogenic markers in 3T3-L1 adipocytes and C3H10T1/2 cells. Relative mRNA expression levels of beige markers and thermogenic markers were measured in 3T3-L1 adipocytes (**a**) and C3H10T1/2 cells (**b**) using RT-PCR. The protein expressions of UCP1, PGC1α, PRDM16, and p-AMPK/AMPK in the 3T3-L1 adipocytes (**c**) and C3H10T1/2 cells (**d**) were analyzed via Western blot analysis. The values are expressed as ±SD (*n* = 3). * *p* < 0.05, ** *p* < 0.01, and *** *p* < 0.001 vs. the control group.

**Figure 5 foods-12-04137-f005:**
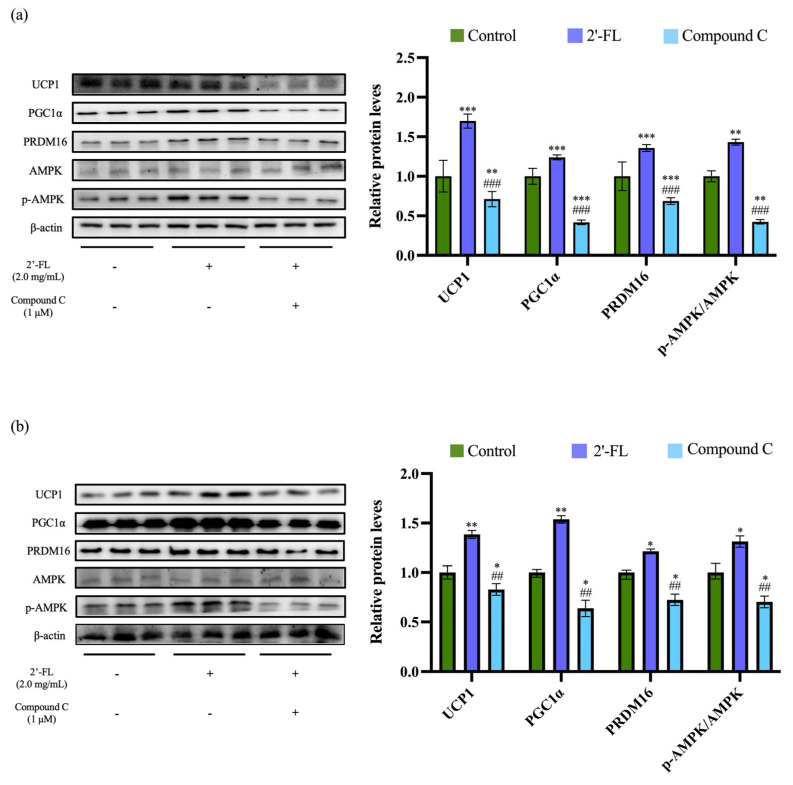
The effect of an AMPK inhibitor (compound C) on 2’-FL-induced thermogenesis in 3T3-L1 and C3H10T1/2 adipocytes. The protein expression levels of UCP1, PGC1α, PRDM16, AMPK, and p-AMPK in 3T3-L1 adipocytes (**a**) and C3H10T1/2 cells (**b**) after compound C treatment. The values are expressed as ± SD (*n* = 3). * *p* < 0.05, ** *p* < 0.01, and *** *p* < 0.001 vs. the control group. ^##^ *p* < 0.01 and ^###^ *p* < 0.001 vs. the 2’-FL group.

**Table 1 foods-12-04137-t001:** Real-time PCR primer sequences.

Gene	Forward Sequence (5′-3′)	Reverse Sequence (5′-3′)
Ucp1	ACTGCCACACCTCCAGTCATT	CTTTGCCTCACTCAGGATTGG
Pgc1α	CCCTGCCATTGTTAAGACC	TGCTGCTGTTCCTGTTTTC
Prdm16	CAGCACGGTGAAGCCATTC	GCGTGCATCCGCTTGTG
Cidea	ATCACAACTGGCCTGGTTACG	TACTACCCGGTGTCCATTTCT
Pparα	TGTCGAATATGTGGGGACAA	AATCTTGCAGCTCCGATCAC
Elov13	GATGGTTCTGGGCACCATCTT	CGTTGTTGTGTGGCATCCTT
Tmem26	GAAACCAGTATTGCAGCACCCAAT	AATATTAGCAGGAGTGTTTGGTGGA
Cd137	TACTACCCGGTGTCCATTTCT	CCTCTGGAGTCACAGAAATGGTGGTA
Cox7a	TTCGAGAACCGAGTAGCTGAGAA	CTGTTGCACCGCCCTTCA
Cyt C	CCAAATCTCCACGGTCTGTTC	ATCAGGGTATCCTCTCCCCAG
Tfam	GAGGCCAGTGTGAACCAGTG	GTAGTGCCTGCTGCTCCTGA
GAPDH	ACCCTTAAGAGGGATGCTGC	CCCAATACGGCCAAATCCGT

## Data Availability

The data used to support the findings of this study are available from the corresponding authors.

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
