# Peer review of "Effect of 2’-Fucosyllactose on Beige Adipocyte Formation in 3T3-L1 Adipocytes and C3H10T1/2 Cells"

_foods, 2023, doi:10.3390/foods12224137_

Round 1

Reviewer 1 Report

Comments and Suggestions for Authors

The study investigated the effect of 2'-fucosyllactose, the most abundant fucosylated human milk oligosaccharides on the browning of white adipose tissue in 3T3-L1 adipocytes and C3H10T1/2 cells.

1- It is necessary to review the entire text, as there is missing space between words, starting with the abstract

2- Line 31: ". Obesity increases the risk of many diseases, including insulin resistance, cardiovascular diseases, and other metabolic disorders 45The prevention and treatment of obesity are mainly divided into three intervention categories—drug, surgical, and behavioral interventions". Please remove the 45 from the sentence and add the period.

3 - Lines "8WAT is mainly distributed in the lower skin and around the internal organs, whereas BAT is mainly distributed in the interscapular, axillary, and paravertebral organs [9].9The..." Please remove the 8 and 9 from the sentence.

4- Standardize references in the text, as there are repeated numbers and numbers outside the bracket.

5- Why did the authors analyze UCP-1 and PGC1 alpha by two different methods such as PCR and western blotting?

6- What is the origin of 2'-FL used in the study? Please add to the methodology.

7- Why did the authors choose doses of 0.5 and 2mg/mL of 2'-FL for the following experiments given that, with the exception of the dose of 8.0, the other doses also did not appear to induce toxicity? 

8- Why did the authors evaluate the effect of 2'-FL in cells and not in mice? Wouldn't it be more visible to evaluate morphologically whether 2'-FL is really influencing the browning of the WAT?

Reviewer 2 Report

Comments and Suggestions for Authors

The main medical and social problem in the twenty-first century has become the problem of overweight and obesity, which has captured more and more countries around the world, resulting in an avalanche of increasing health problems for humanity, economic costs for treating such patients, and reduced quality of life.

The goal of this study was to look at the impact of 2'-fucosyllactose on fat accumulation in 3T3-L1 and C3H10T1/2 cell lines, as well as the pathways that contribute to fat accumulation suppression in cells.

The importance of looking for substances that might influence the excessive buildup of fat in different types of adipose tissue and the transformation of one type of adipose tissue into another is undeniable. The authors used primary fibroblast-like cell lines from mice as an object, which can be used in vitro to transfer the results to people. The findings shed light on how 2'-fucosyllactose affects the activity of genes involved in the control of mitochondrial activity, indicators of white adipose tissue transition into brown adipose tissue, and the AMPK pathway.

The authors' findings allowed us to reach logical conclusions concerning the effect of 2'-fucosyllactose on beige adipocyte development in 3T3-L1 adipocytes and C3H10T1/2 cells.

The literature utilized in the paper is 69% no older than 5 years old, and its usage is logical and validates the relevance of the work, as well as allowing the authors to compare their data with the results of other writers.

There are numerous punctuation problems in the text (lines 31, 42, 44, 47, 70, 72, 288). The authors' data presentation is based on the mean value and the error of the mean value, however this does not allow them to truly convey their beliefs regarding the measure of central tendency and dispersion of the acquired data. In the event of a normal distribution of attributes, data should be presented as mean and standard deviation. The authors do not mention in the statistics section that the obtained data correspond with Gauss's law, thus one must trust on the fact that this point of statistical analysis of the obtained data was made. The occurrence of statistically significant changes in the vitality of fibroblast-like cells 3T3 at an 8 mg/ml dose is doubtful because the scatter of features in the control and the experiment overlap (Figure  1 a).

It is required to state in the table captions that the data are presented as mean +/- standard error of the mean. According to Figure 2, there is also no clear statistically significant difference in cellular fat accumulation (2 b and 2 d) compared to the control, with the scatter of values in the groups overlapping - the significance of the differences there is unlikely to be less than 0.01 or 0.001.

The fact that 3T3-L1 cells are originated from mice means that they may not entirely duplicate human biology. Furthermore, 3T3-L1 cells are a homogeneous cell population that may not accurately represent the variability of adipose tissue in vivo.
